# Enhancing End-to-End Autonomous Driving with Latent World Model

**Yingyan Li**[1,2,3,4]   **Lue Fan**[1,2,3]   **Jiawei He**[1,2,3]   **Yuqi Wang**[1,2,3]   **Yuntao Chen**[1,2,3]

**Zhaoxiang Zhang**[1,2,3,4✉]        **Tieniu Tan**[1,2,3]

[1] Institute of Automation, Chinese Academy of Sciences (CASIA)
[2] New Laboratory of Pattern Recognition (NLPR)
[3] State Key Laboratory of Multimodal Artificial Intelligence Systems (MAIS)
[4] School of Future Technology, University of Chinese Academy of Sciences (UCAS)

## Abstract

In autonomous driving, end-to-end planners directly utilize raw sensor data, enabling them to extract richer scene features and reduce information loss compared to traditional planners. This raises a crucial research question: how can we develop better scene feature representations to fully leverage sensor data in end-to-end driving? Self-supervised learning methods show great success in learning rich feature representations in NLP and computer vision. Inspired by this, we propose a novel self-supervised learning approach using the **LA**tent **W**orld model (**LAW**) for end-to-end driving. LAW predicts future scene features based on current features and ego trajectories. This self-supervised task can be seamlessly integrated into perception-free and perception-based frameworks, improving scene feature learning and optimizing trajectory prediction. LAW achieves state-of-the-art performance across multiple benchmarks, including real-world open-loop benchmark nuScenes, NAVSIM, and simulator-based closed-loop benchmark CARLA. The code is released at `https://github.com/BraveGroup/LAW`.

## 1 Introduction

End-to-end planners (Hu et al., 2022c; Jiang et al., 2023; Prakash et al., 2021; Wu et al., 2022; Hu et al., 2022b; Zhang et al., 2022; Wu et al., 2023) have garnered significant attention due to their distinct advantages over traditional planners. Traditional planners operate on pre-processed outputs from perception modules, such as bounding boxes and trajectories. In contrast, end-to-end planners directly utilize raw sensor data to extract scene features, minimizing information loss. This direct use of sensor data raises an important research question: how can we develop more effective scene feature representations to fully leverage the richness of sensor data in end-to-end driving?

In recent years, self-supervised learning has emerged as a powerful method for extracting comprehensive feature representations from large-scale datasets, particularly in fields like NLP (Devlin, 2018) and computer vision (He et al., 2022). Building on this success, we aim to enrich scene feature learning and further improve end-to-end driving performance through self-supervised learning. Traditional self-supervised methods in computer vision(He et al., 2022; Chen et al., 2020b) often focus on static, single-frame images. However, autonomous driving relies on continuous video input, so effectively using temporal data is crucial. Temporal self-supervised tasks, such as future prediction (Han et al., 2019; 2020), have shown promise. Traditional future prediction tasks often overlook the impact of ego actions, which play a crucial role in shaping the future in autonomous driving.

Considering the critical role of ego actions, we propose utilizing a world model to predict future states based on the current state and ego actions. While several image-based driving world models (Wang et al., 2023b; Hu et al., 2023a; Jia et al., 2023a) exist, they exhibit inefficiencies in enhancing scene

---

[1]Email: liyingyan2021@ia.ac.cn

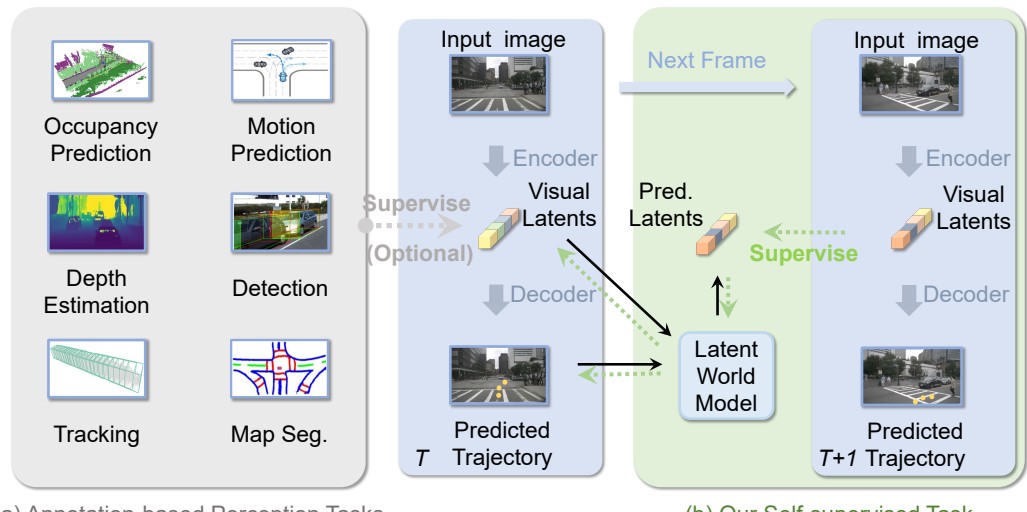

Figure 1: **The illustration of our self-supervised method.** Traditional methods utilize perception annotations to assist with scene feature learning. In contrast, our self-supervised approach uses temporal information to guide feature learning. Pred.: predicted. Seg.: segmentation. The blue part indicates the pipeline of an end-to-end planner.

feature representations due to their reliance on diffusion models, which may take several seconds to generate images of a future scene. To address this limitation, we introduce a latent world model designed to predict future latent features directly from the current latent features and ego actions, as depicted in Fig. 1. Specifically, given images, a visual encoder extracts scene features (current state), which are then fed into an action decoder to predict ego trajectory. Based on the current state and action, the latent world model predicts the scene feature of the future frame. During training, the predicted future features are supervised using the extracted features from the future frame. By supervising the predicted future feature, this self-supervised method jointly optimizes the current scene feature learning and ego trajectory prediction.

After introducing the concept of the latent world model, we turn our attention to its universality across various end-to-end autonomous driving frameworks. In end-to-end autonomous driving, the frameworks can generally be categorized into two types: perception-free and perception-based. Perception-free approaches (Toromanoff et al., 2020; Chen et al., 2020a; Wu et al., 2022) bypass explicit perception tasks, relying solely on trajectory supervision. Prior work Wu et al. (2022) in this category typically extracts perspective-view features to predict future trajectory. In contrast, perception-based approaches (Prakash et al., 2021; Hu et al., 2022c; Jiang et al., 2023; Hu et al., 2022b) incorporate perception tasks, such as detection, tracking, and map segmentation, to guide scene feature learning. These methods generally use BEV feature maps as a unified representation for these perception tasks. Our latent world model accommodates both frameworks. It can either predict perspective-view features in the perception-free setting or predict BEV features in the perception-based setting, showcasing its universality across different autonomous driving paradigms.

Experiments show that our latent world model enhances performance in both perception-free and perception-based frameworks. Furthermore, we achieve state-of-the-art performance on multiple benchmarks, including the real-world open-loop datasets nuScenes (Caesar et al., 2020) and NAVSIM (Dauner et al., 2024) (based on nuPlan (Caesar et al., 2021)), as well as the simulator-based closed-loop CARLA benchmark (Dosovitskiy et al., 2017). These results underscore the efficacy of our approach and highlight the potential of self-supervised learning to advance end-to-end autonomous driving research. In summary, our contributions are threefold:

- **Future prediction by latent world model:** We introduce the *LA*tent *W*orld model (*LAW*) to predict future scene latents from current scene latents and ego trajectories. This self-

supervised task jointly enhances scene representation learning and trajectory prediction in end-to-end driving.

- **Cross-framework universality:** *LAW* demonstrates universality across various common autonomous driving paradigms. It can either predict perspective-view features in the perception-free framework or predict BEV features in the perception-based framework.

- **State-of-the-art performance:** Our self-supervised approach achieves state-of-the-art results on the real-world open-loop nuScenes, NAVSIM, and the simulator-based close-loop CARLA benchmark.

## 2 RELATED WORKS

### 2.1 END-TO-END AUTONOMOUS DRIVING

We divide end-to-end autonomous driving methods (Hu et al., 2022c; Jiang et al., 2023; Renz et al., 2022; Toromanoff et al., 2020; Tian et al., 2024; Hwang et al., 2024; Pan et al., 2024; Wang et al., 2024a;b) into two categories, perception-based methods and perception-free methods, depending on whether performing perception tasks. Perception-based methods (Casas et al., 2021; Prakash et al., 2021; Jaeger et al., 2023; Shao et al., 2023; Hu et al., 2022b; Sadat et al., 2020) perform multiple perception tasks simultaneously, such as detection (Li et al., 2022b; Huang et al., 2021; Li et al., 2022a; 2024a), tracking (Zhou et al., 2020; Wang et al., 2021), map segmentation (Hu et al., 2022c; Jiang et al., 2023) and occupancy prediction (Wang et al., 2023a; Huang et al., 2023). As a representative, UniAD (Hu et al., 2022c) integrates multiple modules to support goal-driven planning. VAD (Jiang et al., 2023) explores vectorized scene representation for planning purposes.

Perception-free end-to-end methods (Toromanoff et al., 2020; Chen et al., 2020a; Zhang et al., 2021; Wu et al., 2022) present a promising direction as they avoid utilizing a large number of perception annotations. Early perception-free end-to-end methods (Zhang et al., 2021; Toromanoff et al., 2020) primarily relied on reinforcement learning. For instance, MaRLn (Toromanoff et al., 2020) designed a reinforcement learning algorithm based on implicit affordances, while LBC (Chen et al., 2020a) trained a reinforcement learning expert using privileged (ground-truth perception) information. Using trajectory data generated by the reinforcement learning expert, TCP (Wu et al., 2022) combined a trajectory waypoint branch with a direct control branch, achieving good performance. However, perception-free end-to-end methods often suffer from inadequate scene representation capabilities. Our work aims to address this issue through the latent world model.

### 2.2 WORLD MODEL IN AUTONOMOUS DRIVING

Existing world models in autonomous driving can be categorized into two types: image-based world models and occupancy-based world models. Image-based world models (Hu et al., 2022a; Wang et al., 2023b; Hu et al., 2023a) aim to enrich the autonomous driving dataset through generative approaches. GAIA-1 (Hu et al., 2023a) is a generative world model that utilizes video, text, and action inputs to create realistic driving scenarios. MILE (Hu et al., 2022a) produces urban driving videos by leveraging 3D geometry as an inductive bias. Drive-WM (Wang et al., 2023b) utilizes a diffusion model to predict future images and then plans based on these predicted images. Copilot4D (Zhang et al., 2023) tokenizes sensor observations with VQVAE (Van Den Oord et al., 2017) and then predicts the future via discrete diffusion. Another category involves occupancy-based world models (Zheng et al., 2023; Min et al., 2024). OccWorld (Zheng et al., 2023) and DriveWorld (Min et al., 2024) use the world model to predict the occupancy, which requires occupancy annotations. On the contrary, our proposed latent world model requires no manual annotations.

## 3 PRELIMINARY

**Vision-based End-to-end Autonomous Driving** In the task of end-to-end autonomous driving, the objective is to estimate the future trajectory of the ego vehicle in the form of waypoints. Formally, let $\mathbf{I}_t = \{\mathbf{I}_t^1, \mathbf{I}_t^2, \ldots, \mathbf{I}_t^N\}$ be the set of $N$ surrounding multi-view images captured at time step $t$. We expect the model to predict a sequence of waypoints $\mathbf{W}_t = \{\mathbf{w}_t^1, \mathbf{w}_t^2, \ldots, \mathbf{w}_t^M\}$, where each

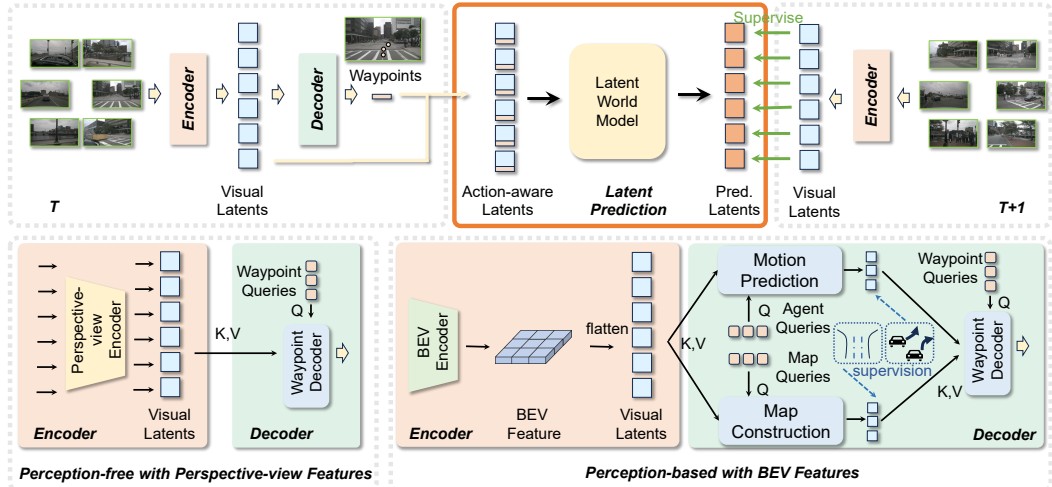

Figure 2: **The Overall Framework.** The encoder extracts visual latents from the images, while the decoder predicts waypoints based on these latents. Our latent world model predicts future visual latents using the visual latents and waypoints of the current frame. During training, the predicted visual latents are supervised by the extracted latents from the future frame. The latent world model is compatible with both perception-free and perception-based frameworks, which differ in their encoder and decoder structures. In the perception-based framework, the supervision icon indicates that map annotations supervise the output of the map construction module, while the agent's future trajectory supervises the output of the motion prediction module. Pred.: predicted.

waypoint $\mathbf{w}_t^i = (x_t^i, y_t^i)$ represents the predicted BEV position of the ego vehicle at time step $t + i$. $M$ represents the number of future positions of the ego vehicle that the model aims to predict.

**World Model** A world model aims to predict future states based on the current state and action. In the autonomous driving task, let $\mathbf{S}_t$ represent the states at time step $t$, $\mathbf{W}_t = \{\mathbf{w}_t^1, \mathbf{w}_t^2, \ldots, \mathbf{w}_t^M\}$ denote the sequence of predicted waypoints by the planner, the world model predicts state $\mathbf{S}_{t+1}$ at time step $t + 1$ using $\mathbf{S}_t$ and $\mathbf{W}_t$.

## 4 METHODOLOGY

Our methodology is composed of three key components: i) Latent World Model: We utilize the latent world model to realize the self-supervised task. This model takes two inputs: latent features extracted by the visual encoder and waypoints predicted by the waypoint decoder. This task is compatible with two common frameworks. ii) Perception-free framework with perspective-view latents: it consists of the perspective-view encoder and perception-free decoder within this framework. iii) Perception-based framework with BEV latents: it contains the BEV encoder and perception-based decoder within this framework.

### 4.1 LATENT WORLD MODEL

In this section, we utilize the latent world model to predict the visual latents of the future frame based on the current visual latents and waypoints.

**Visual Latents and Waypoints Extraction** The visual encoder processes the images from the current $t$ time step to produce the corresponding visual latent feature set

$$\mathbf{V}_t = \{\mathbf{v}_t^1, \mathbf{v}_t^2, \ldots, \mathbf{v}_t^L\},$$

where $L$ denotes the number of feature vectors, and each vector $\mathbf{v}_t^i \in \mathbb{R}^D$, with $D$ representing the number of feature channels. These feature vectors can be derived from various sources, such as a flattened image feature map or a flattened BEV feature map. Based on the $\mathbf{V}_t$, the waypoint decoder

predicts the waypoints $\mathbf{W}_t = \{\mathbf{w}_t^1, \mathbf{w}_t^2, \ldots, \mathbf{w}_t^M\}$. $M$ represents the number of waypoints, where each waypoint $\mathbf{w}_t^i = (x_t^i, y_t^i)$.

**Action-aware Latents Construction** We produce action-aware latents by integrating visual latents and waypoints. The action-aware latents are then used as input to the latent world model. Let $M$ represent the number of waypoints, with each waypoint $\mathbf{w}_t^i = (x_t^i, y_t^i)$. We first reshape $\mathbf{W}_t$, which has the shape $[M, 2]$, into a one-dimensional vector $\widetilde{\mathbf{w}}_t \in \mathbb{R}^{2M}$. Then, we concatenate each visual latent $\mathbf{v}_t^i$ with the waypoint vector $\widetilde{\mathbf{w}}_t$ along the feature channel dimension. The resulting concatenated vector is passed through an MLP to produce the action-aware latent $\mathbf{a}_t^i$, which has the same shape as $\mathbf{v}_t^i$. Formally, the action-aware latent for the $i$-th feature vector is expressed as

$$\mathbf{a}_t^i = \text{MLP}([\mathbf{v}_t^i, \widetilde{\mathbf{w}}_t]), \tag{1}$$

where $[\cdot, \cdot]$ denotes the concatenating operation. The full set of action-aware latents is denoted as $\mathbf{A}_t = \{\mathbf{a}_t^1, \mathbf{a}_t^2, \ldots, \mathbf{a}_t^L\}$, where $L$ denotes the number of feature vectors.

**Future Latent Prediction** The latent world model utilizes the action-aware latents to predict the future visual latents as follows. Given $\mathbf{A}_t$, we predict visual latents $\hat{\mathbf{V}}_{t+1}$ of the frame $t + 1$ by the latent world model:

$$\hat{\mathbf{V}}_{t+1} = \text{LatentWorldModel}(\mathbf{A}_t). \tag{2}$$

The network architecture of the latent world model consists of transformer blocks. Each block contains a self-attention and feed-forward module. The self-attention is performed across the latent feature vectors.

**Future Latent Supervision** During training, we extract the visual latents $\mathbf{V}_{t+1} = \{\mathbf{v}_{t+1}^1, \ldots, \mathbf{v}_{t+1}^L\}$ from the images of frame $t + 1$, which are used as the ground truth to supervise the $\hat{\mathbf{V}}_{t+1}$ using a Mean Squared Error (MSE) loss function as

$$\mathcal{L}_{\text{latent}} = \frac{1}{L} \sum_{i=1}^{L} \|\hat{\mathbf{v}}_{t+1}^i - \mathbf{v}_{t+1}^i\|_2. \tag{3}$$

The latent world model is compatible with both perception-free and perception-based frameworks. In the following sections, we detail the implementation of these two frameworks.

## 4.2 Perception-free Framework with Perspective-view Latents

First, we introduce our perception-free framework. Previous perception-free frameworks (Wu et al., 2022) typically employ a perspective-view encoder for visual latent extraction and a perception-free decoder for waypoint prediction. Our framework is built upon this established paradigm.

**Perspective-view Encoder** In the perspective-view encoder, we produce visual latents based on multi-view images. Initially, multi-view images are processed by an image backbone to obtain their corresponding image features. Following PETR (Liu et al., 2022), we generate 3D position embeddings for these image features, which are then added to the image features to uniquely distinguish each feature vector. The enriched image features are denoted as $\mathbf{F} = \{\mathbf{f}^1, \mathbf{f}^2, \ldots, \mathbf{f}^N\}$, where $N$ represents the number of views. The shape of $\mathbf{f}^i$ is $[H, W, D]$, where $H, W$ represents the height and width of the image feature map and $D$ is the number of feature channels.

To encode the image features into high-level visual latents suitable for planning, we apply a view attention mechanism. To be specific, for $N$ views, there are $N$ corresponding learnable view queries $\mathbf{Q}_{\text{view}} = \{\mathbf{q}_{\text{view}}^1, \mathbf{q}_{\text{view}}^2, \ldots, \mathbf{q}_{\text{view}}^N\}$. Each view query $\mathbf{q}_{\text{view}}^i$ undergoes a cross-attention with its corresponding image feature $\mathbf{f}^i$, resulting in $N$ visual latent $\mathbf{V}_{\text{pf}} = \{\mathbf{v}_{\text{pf}}^1, \mathbf{v}_{\text{pf}}^2, \ldots, \mathbf{v}_{\text{pf}}^N\}$, where the subscript "pf" stands for perception-free. Formally,

$$\mathbf{v}_{\text{pf}}^i = \text{CrossAttention}(\mathbf{q}_{\text{view}}^i, \mathbf{f}^i, \mathbf{f}^i), \tag{4}$$

where $\mathbf{f}^i$ serves as the key and value of the cross attention. $\mathbf{V}_{\text{pf}}$ are then used as input for the perception-free decoder, which we describe next.

**Perception-free Decoder** The perception-free decoder deocdes waypoint from $\mathbf{V}_{\text{pf}}$. Specifically, we initialize $M$ waypoint queries, $\mathbf{Q}_{\text{wp}} = \{\mathbf{q}_{\text{wp}}^1, \mathbf{q}_{\text{wp}}^2, \ldots, \mathbf{q}_{\text{wp}}^M\}$, where each query is a learnable embedding. These waypoint queries interact with $\mathbf{V}_{\text{pf}}$ through a cross-attention mechanism. The

updated waypoint queries are then passed through an MLP head to predict the waypoints $\mathbf{W} = \{\mathbf{w}^1, \mathbf{w}^2, \ldots, \mathbf{w}^M\}$, which is formulated as

$$\mathbf{W} = \text{MLP}(\text{CrossAttention}(\mathbf{Q}_{\text{wp}}, \mathbf{V}_{\text{pf}}, \mathbf{V}_{\text{pf}})). \tag{5}$$

**Perception-free Supervision** In the perception-free framework, we rely solely on ground truth waypoints for supervision, as no additional perception annotations are provided. We employ an L1 loss to measure the discrepancy between the predicted waypoints $\mathbf{W}$ and the ground truth waypoints $\tilde{\mathbf{W}} = \{\tilde{\mathbf{w}}^1, \tilde{\mathbf{w}}^2, \ldots, \tilde{\mathbf{w}}^M\}$ as

$$\mathcal{L}_{\text{waypoint}} = \frac{1}{M} \sum_{j=1}^{M} \|\mathbf{w}_t^j - \tilde{\mathbf{w}}_t^j\|_1, \tag{6}$$

Thus, the final loss for the perception-free framework is

$$\mathcal{L}_{\text{pf}} = \mathcal{L}_{\text{latent}} + \mathcal{L}_{\text{waypoint}}. \tag{7}$$

### 4.3 PERCEPTION-BASED FRAMEWORK WITH BEV LATENTS

Our latent world model is also compatible with perception-based frameworks, which commonly utilize BEV feature maps for perception tasks. We adhere to this paradigm and the perception-based framework is composed of two key components: a BEV encoder and a perception-based decoder. The BEV encoder generates BEV feature maps from images, while the perception-based decoder uses these maps for perception tasks such as motion prediction and map construction. The final waypoints are then predicted based on the outputs of these perception tasks.

**BEV Encoder** We follow Li et al. (2022b) to encode the BEV feature map. First, we encode the image features using a backbone network. Then, a set of BEV queries projects these image features into BEV features. The resulting BEV feature map is flattened into a shape of $[K, D]$, where $K$ represents the number of feature vectors in the BEV feature map and $D$ is the number of feature channels. The flattened features are denoted as $\mathbf{V}_{\text{pb}} = \{\mathbf{v}_{\text{pb}}^1, \mathbf{v}_{\text{pb}}^2, \ldots, \mathbf{v}_{\text{pb}}^K\}$, where the subscript "pb" refers to perception-based.

**Perception-based Decoder** Following Jiang et al. (2023), the decoder predicts waypoints with the help of perception tasks, namely motion prediction and map construction. For motion prediction, $N_{\text{agent}}$ learnable queries interact with $\mathbf{V}_{\text{pb}}$ via cross-attention to generate agent features $\mathbf{F}_{\text{agent}}$. $\mathbf{F}_{\text{agent}}$ are then used to predict agent trajectories. Similarly, for map construction, $N_{\text{map}}$ learnable queries perform cross-attention with $\mathbf{V}_{\text{pb}}$ to extract map features $\mathbf{F}_{\text{map}}$. $\mathbf{F}_{\text{map}}$ are then used to predict map vectors. Finally, the learnable waypoint queries perform cross-attention with $\mathbf{F}_{\text{agent}}$ and $\mathbf{F}_{\text{map}}$. The output is then passed through an MLP head to predict the waypoints.

**Perception-based Supervision** The perception-based framework uses the same waypoint supervision as in equation 6. In addition, it includes losses from perception tasks as

$$\mathcal{L}_{\text{perception}} = \mathcal{L}_{\text{agent}} + \mathcal{L}_{\text{map}}. \tag{8}$$

Here, $\mathcal{L}_{\text{agent}}$ is s the loss for motion prediction and $\mathcal{L}_{\text{map}}$ is the loss for map construction, as defined in (Jiang et al., 2023). The final loss for the perception-based framework is

$$\mathcal{L}_{\text{pb}} = \mathcal{L}_{\text{latent}} + \mathcal{L}_{\text{waypoint}} + \mathcal{L}_{\text{perception}}. \tag{9}$$

## 5 EXPERIMENTS

### 5.1 BENCHMARKS

**nuScenes (Caesar et al., 2020)** The nuScenes dataset contains 1,000 driving scenes. In line with previous works (Hu et al., 2022b; 2023b; Jiang et al., 2023), we use L2 displacement error and collision rate as comprehensive metrics to evaluate planning performance. L2 displacement error measures the L2 distance between the predicted and ground truth trajectories, while collision rate quantifies the frequency of collisions with other objects along the predicted trajectory.

**NAVSIM (Dauner et al., 2024)** We conducted further experiments using the NAVSIM benchmark, as the nuScenes dataset proved to be overly simplistic. The NAVSIM dataset (Dauner et al., 2024)

is built on OpenScene (Contributors, 2023), which provides 120 hours of driving logs condensed from the nuPlan dataset (Caesar et al., 2021). NAVSIM enhances OpenScene by resampling the data to reduce the occurrence of simple scenarios, such as straight-line driving. As a result, traditional ego status modeling becomes inadequate under the NAVSIM benchmark. NAVSIM evaluates model performance using the predictive driver model score (PDMS), which is calculated based on five factors: no at-fault collision (NC), drivable area compliance (DAC), time-to-collision (TTC), comfort (Comf.) and ego progress (EP).

**CARLA (Dosovitskiy et al., 2017)** Closed-loop evaluation is essential to autonomous driving as it constantly updates the sensor inputs based on the driving actions. For the closed-loop benchmark, the training dataset is collected from the CARLA (Dosovitskiy et al., 2017) simulator (version 0.9.10.1) using the teacher model Roach (Zhang et al., 2021) following (Wu et al., 2022; Jia et al., 2023b), resulting in 189K frames. We use the widely-used Town05 Long benchmark (Jia et al., 2023b; Shao et al., 2022; Hu et al., 2022a) to assess the closed-loop driving performance. For metric, Route Completion (RC) represents the percentage of the route completed by the autonomous driving model. Infraction Score (IS) quantifies the number of infractions as well as violations of traffic rules. A higher Infraction Score indicates better adherence to safe driving practices. Driving Score (DS) is the primary metric used to evaluate overall performance. It is calculated as the product of Route Completion and Infraction Score.

## 5.2 Implementation Details

**nuScenes Benchmark** We implement both perception-free and perception-based frameworks. In the perception-free framework, Swin-Transformer-Tiny (Swin-T)(Liu et al., 2021) is used as the backbone. Input images are resized to 800×320. We adopt a Cosine Annealing learning rate schedule(Loshchilov & Hutter, 2016), starting at 5e-5. The model is trained using the AdamW optimizer (Loshchilov & Hutter, 2017) with a weight decay of 0.01, batch size 8, and 12 epochs across 8 A6000 GPUs. For the perception-based framework, following Jiang et al. (2023), we train the model in two stages. In the first stage, we train the encoder and perception head using only perception loss for 48 epochs. In the second stage, we introduce waypoint and latent prediction losses for training another 12 epochs. The network architecture of the latent world model utilizes deformable self-attention for improved convergence.

**NAVSIM Benchmark** The perception-free framework is implemented on NAVSIM. Specifically, We employ a ResNet-34 backbone, training for 20 epochs in line with Prakash et al. (2021) to ensure a fair comparison. Input images are resized to 640×320. The Adam optimizer is used with a learning rate of 1e-4 and a batch size of 32.

**CARLA Benchmark** We follow Wu et al. (2022) to implement a perception-free framework on CARLA. To be specific, we use ResNet-34 as the backbone and employ the TCP head (Wu et al., 2022) as in Jia et al. (2023b). Input images are resized to 900×256. The Adam optimizer is used with a learning rate of 1e-4 and weight decay of 1e-7. The model is trained for 60 epochs with a batch size of 128. After 30 epochs, the learning rate is halved.

## 5.3 Comparison with State-of-the-art Methods

For the nuScenes benchmark, we compare our proposed framework with several state-of-the-art methods, including BEV-Planner (Li et al., 2024b) and VAD (Jiang et al., 2023). The results are summarized in Table 1. Our perception-free framework demonstrates competitive performance, while the perception-based framework achieves state-of-the-art results in both L2 displacement and collision rates. For the NAVSIM benchmark, detailed in Table 2, our method achieves state-of-the-art results in overall PDMS. For the CARLA benchmark, as shown in Table 3, our proposed method outperforms all existing methods. Notably, our perception-free approach surpasses previous leading methods such as ThinkTwice (Jia et al., 2023c) and DriveAdapter (Jia et al., 2023b), which incorporate extensive supervision from depth estimation, semantic segmentation, and map segmentation.

## 5.4 Ablation Study

All experiments are conducted within the perception-free framework unless otherwise specified.

Table 1: **Performance on the nuScenes (Caesar et al., 2020)**. The overall collision results are computed by the traditional computation way used in Jiang et al. (2023). ‡: The collision results are computed by the way in Li et al. (2024b). We do not utilize the historical ego status information.

| Method | **L2** (m) ↓ | | | | **Collision** (%) ↓ | | | |
|---|---|---|---|---|---|---|---|---|
| | 1s | 2s | 3s | Avg. | 1s | 2s | 3s | Avg. |
| NMP (Zeng et al., 2019) | - | - | 2.31 | - | - | - | 1.92 | - |
| SA-NMP (Zeng et al., 2019) | - | - | 2.05 | - | - | - | 1.59 | - |
| FF (Hu et al., 2021) | 0.55 | 1.20 | 2.54 | 1.43 | 0.06 | 0.17 | 1.07 | 0.43 |
| EO (Khurana et al., 2022) | 0.67 | 1.36 | 2.78 | 1.60 | 0.04 | 0.09 | 0.88 | 0.33 |
| ST-P3 (Hu et al., 2022b) | 1.33 | 2.11 | 2.90 | 2.11 | 0.23 | 0.62 | 1.27 | 0.71 |
| UniAD (Hu et al., 2022c) | 0.48 | 0.96 | 1.65 | 1.03 | 0.05 | 0.17 | 0.71 | 0.31 |
| VAD (Jiang et al., 2023) | 0.41 | 0.70 | 1.05 | 0.72 | 0.07 | 0.17 | 0.41 | 0.22 |
| BEV-Planner (Li et al., 2024b) | 0.30 | 0.52 | 0.83 | 0.55 | 0.10‡ | 0.37‡ | 1.30‡ | 0.59‡ |
| LAW (perception-free) | 0.26 | 0.57 | 1.01 | 0.61 | 0.14 | 0.21 | 0.54 | 0.30 |
| LAW (perception-based) | 0.24 | 0.46 | 0.76 | 0.49 | 0.08 | 0.10 | 0.39 | 0.19 |

Table 2: **Performance on NAVSIM test set.** NC: no at-fault collision. DAC: drivable area compliance. TTC: time-to-collision. Comf.: comfort. EP: ego progress. PDMS: the predictive driver model score. LAW is in the perception-free setting.

| Method | NC ↑ | DAC ↑ | TTC ↑ | Comf. ↑ | EP ↑ | PDMS ↑ |
|---|---|---|---|---|---|---|
| Human | 100 | 100 | 100 | 99.9 | 87.5 | 94.8 |
| Constant Velocity | 69.9 | 58.8 | 49.3 | 100 | 49.3 | 21.6 |
| Ego Status MLP | 93.0 | 77.3 | 83.6 | 100 | 62.8 | 65.6 |
| TransFuser (Prakash et al., 2021) | 97.7 | 92.8 | 92.8 | 100 | 79.2 | 84.0 |
| UniAD (Hu et al., 2022c) | 97.8 | 91.9 | 92.9 | 100 | 78.8 | 83.4 |
| PARA-Drive (Weng et al., 2024) | 97.9 | 92.4 | 93.0 | 99.8 | 79.3 | 84.0 |
| LAW | 96.4 | 95.4 | 88.7 | 99.9 | 81.7 | 84.6 |

Table 3: **Performance on Town05 Long benchmark on CARLA.** Expert: Imitation learning from the driving trajectories of a privileged expert. Seg.: semantic segmentation. Map.: BEV map segmentation. Dep.: depth estimation. Det.: 3D object detection. *Latent Prediction*: our proposed self-supervised task. RC: route completion. IS: infraction score. DS: driving score. LAW is in the perception-free setting.

| Method | Supervision | RC↑ | IS↑ | DS↑ |
|---|---|---|---|---|
| CILRS (Codevilla et al., 2019) | Expert | 10.3±0.0 | 0.75±0.05 | 7.8±0.3 |
| LBC (Chen et al., 2020a) | Expert | 31.9±2.2 | 0.66±0.02 | 12.3±2.0 |
| Transfuser (Prakash et al., 2021) | Expert, Dep., Seg., Map., Det. | 47.5±5.3 | 0.77±0.04 | 31.0±3.6 |
| Roach (Zhang et al., 2021) | Expert | 96.4±2.1 | 0.43±0.03 | 41.6±1.8 |
| LAV (Chen & Krähenbühl, 2022) | Expert, Seg., Map., Det. | 69.8±2.3 | 0.73±0.02 | 46.5±2.3 |
| TCP (Wu et al., 2022) | Expert | 80.4±1.5 | 0.73±0.02 | 57.2±1.5 |
| MILE (Hu et al., 2022a) | Expert, Map., Det. | 97.4±0.8 | 0.63±0.03 | 61.1±3.2 |
| ThinkTwice (Jia et al., 2023c) | Expert, Dep., Seg., Det. | 95.5±2.0 | 0.69±0.05 | 65.0±1.7 |
| DriveAdapter (Jia et al., 2023b) | Expert, Map., Det. | 94.4±- | 0.72±- | 65.9±- |
| Interfuser (Shao et al., 2022) | Expert, Map., Det. | 95.0±2.9 | - | 68.3±1.9 |
| LAW | Expert, *Latent Prediction* | 97.8±0.9 | 0.72±0.03 | 70.1±2.6 |

**Ablation Study on Latent World Model** In this ablation study, we assess the effectiveness of our proposed latent world model. For the nuScenes benchmark, the results are shown in Table 1. We ablate the latent prediction task in both the perception-free and perception-based frameworks, and further investigate the contribution of each input to the latent world model. The findings demonstrate that accurate future latent predictions depend on incorporating driving actions, supporting the validity of the latent world model. We also present ablation studies on NAVSIM and CARLA, as detailed in Table 5. In NAVSIM, we observed a significant improvement in PDMS, mainly driven by the

Table 4: **Effectiveness of latent prediction on nuScenes benchmark.** The latent world model receives two types of inputs: visual latents and predicted trajectory. No input refers to not utilizing the world model. Pred.: predicted. Traj.: trajectory. Avg.: average.

| Framework | Input of Latent World Model | | L2 (m) ↓ | | | | Collision (%) ↓ | | | |
| --- | --- | --- | --- | --- | --- | --- | --- | --- | --- | --- |
| | Visual Latent | Pred. Traj. | 1s | 2s | 3s | Avg. | 1s | 2s | 3s | Avg. |
| Perception-free | - | - | 0.32 | 0.67 | 1.14 | 0.71 | 0.20 | 0.30 | 0.73 | 0.41 |
| | ✓ | - | 0.30 | 0.64 | 1.12 | 0.68 | 0.18 | 0.27 | 0.66 | 0.37 |
| | ✓ | ✓ | 0.26 | 0.57 | 1.01 | 0.61 | 0.14 | 0.21 | 0.54 | 0.30 |
| Perception-based | - | - | 0.30 | 0.52 | 0.80 | 0.54 | 0.09 | 0.17 | 0.48 | 0.25 |
| | ✓ | - | 0.27 | 0.49 | 0.80 | 0.52 | 0.08 | 0.12 | 0.42 | 0.21 |
| | ✓ | ✓ | 0.24 | 0.46 | 0.76 | 0.49 | 0.08 | 0.10 | 0.39 | 0.19 |

Table 5: **Ablation study on latent prediction on NAVSIM and CARLA benchmark.** NC: no at-fault collision. DAC: drivable area compliance. TTC: time-to-collision. Comf.: comfort. EP: ego progress. PDMS: the predictive driver model score. RC: route Completion. IS: infraction Score. DS: driving Score. LAW is in the perception-free setting.

| Latent Prediction | NAVSIM | | | | | | CARLA | | |
| --- | --- | --- | --- | --- | --- | --- | --- | --- | --- |
| | NC ↑ | DAC ↑ | TTC ↑ | Comf. ↑ | EP ↑ | PDMS ↑ | RC ↑ | IS ↑ | DS ↑ |
| ✗ | 94.4 | 89.4 | 84.8 | 100.0 | 75.1 | 77.5 | 98.6±0.8 | 0.68±0.02 | 67.9±2.1 |
| ✓ | 96.4 | 95.4 | 88.7 | 99.9 | 81.7 | 84.6 | 97.8±0.9 | 0.72±0.03 | 70.1±2.6 |

enhancements in drivable area compliance (DAC) and Ego progress (EP) metrics. This suggests that our self-supervised task effectively enhances the quality of the driving trajectory. Similarly, in CARLA, we observed notable improvements in the Driving Score.

**The Time Horizon of Latent World Model** In this experiment, the world model predicts latent features at three distinct future time horizons: 0.5 seconds, 1.5 seconds, and 3.0 seconds. This corresponds to the first, third, and sixth future frames from the current frame, given that keyframes occur every 0.5 seconds in the nuScenes dataset. The results, displayed in Table 6, show that the model achieves the best performance at the 1.5-second horizon. In comparison, the 0.5-second interval typically presents scenes with minimal changes, providing insufficient dynamic content to improve feature learning. In contrast, the 3.0-second interval often presents scenes that may significantly differ from the current frame, making accurate future predictions more challenging. Moreover, we observe that predicting latents 10 seconds into the future completely diminishes the gains provided by the world model, suggesting that predicting features too far into the future is ineffective. This conclusion aligns with observations from MAE (He et al., 2022), where both excessively low and high mask ratios negatively impact the ability of the network.

Table 6: **Different time horizons for latent prediction.** The time intervals of 0.5, 1.5, 3.0, and 10.0 seconds correspond to the first, third, sixth, and twentieth future frames from the current frame, as keyframes occur every 0.5 seconds in the nuScenes dataset.

| Time Horizon | L2 (m) ↓ | | | | Collision (%) ↓ | | | |
| --- | --- | --- | --- | --- | --- | --- | --- | --- |
| | 1s | 2s | 3s | Avg. | 1s | 2s | 3s | Avg. |
| 0.5s | 0.26 | 0.57 | 1.01 | 0.61 | 0.14 | 0.21 | 0.54 | 0.30 |
| 1.5s | 0.26 | 0.54 | 0.93 | 0.58 | 0.14 | 0.17 | 0.45 | 0.25 |
| 3.0s | 0.28 | 0.59 | 1.01 | 0.63 | 0.13 | 0.20 | 0.48 | 0.27 |
| 10.0s | 0.33 | 0.67 | 1.16 | 0.72 | 0.24 | 0.28 | 0.76 | 0.43 |

**Network Architecture of Latent World Model** To validate the impact of the network architecture of the latent world model, we conduct experiments as shown in Table 7. Firstly, it is evident that a single-layer neural network, represented as Linear Projection, is not adequate for fulfilling the functions of the world model, resulting in significantly degraded performance. The two-layer MLP shows considerable improvement in performance. However, it lacks the capability to facilitate interactions among different latent vectors. Therefore, we use the stacked transformer blocks as our

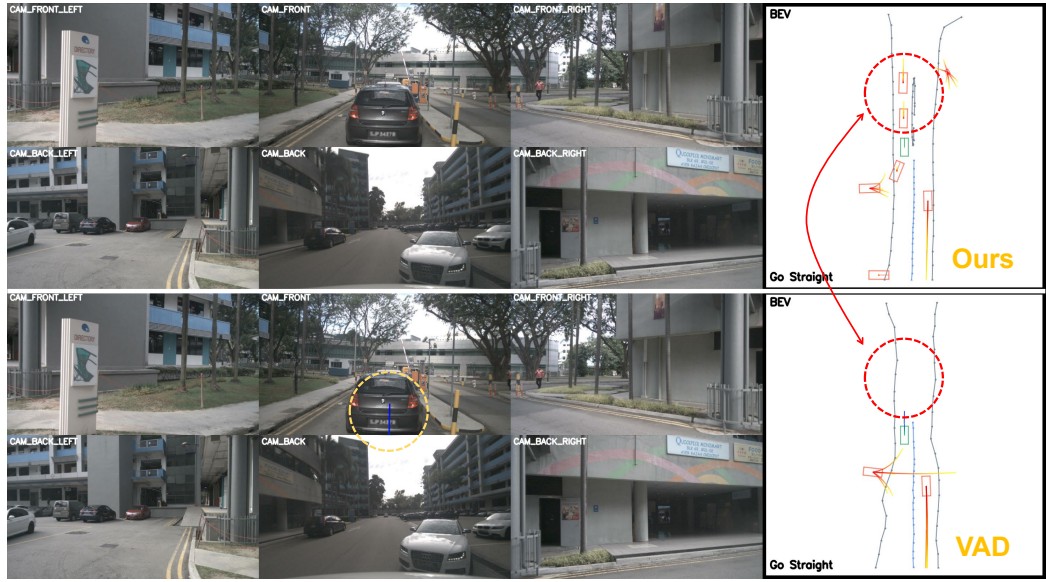

Figure 3: **Visualization.** This figure compares LAW in the perception-based setting with VAD (Jiang et al., 2023). On the right side of this figure, we display the results of ego trajectory prediction, agent motion prediction, and map construction in BEV. As indicated by the red circles, our method captures more crucial scene information, which VAD overlooks. Consequently, VAD predicts a forward trajectory that results in a rear-end collision, as highlighted by the yellow circles.

default network architecture, which achieves the best results among the tested architectures. This indicates that interactions between feature vectors from different positions are important.

Table 7: **Different network architecture of the latent world model.** Linear Projection means a single-layer network.

| Architecture | L2 (m) ↓ | | | | Collision (%) ↓ | | | |
|---|---|---|---|---|---|---|---|---|
| | 1s | 2s | 3s | Avg. | 1s | 2s | 3s | Avg. |
| Linear Projection | 0.31 | 0.65 | 1.14 | 0.70 | 0.26 | 0.34 | 0.66 | 0.42 |
| Two-layer MLP | 0.27 | 0.58 | 1.07 | 0.64 | 0.17 | 0.23 | 0.59 | 0.33 |
| Transformer Blocks | 0.26 | 0.57 | 1.01 | 0.61 | 0.14 | 0.21 | 0.54 | 0.30 |

## 5.5 VISUALIZATION

Figure 3 compares the results of LAW in the perception-based setting with VAD (Jiang et al., 2023). Leveraging our latent world model, our approach acquires more comprehensive scene representations.

## 6 CONCLUSION

In conclusion, we present the latent world model to predict future features from current features and ego trajectories, which is a novel self-supervised learning method for end-to-end autonomous driving. This method jointly enhances scene representation learning and ego trajectory prediction. Our approach demonstrates universality by accommodating both perception-free and perception-based frameworks, predicting perspective-view features and BEV features respectively. We achieve state-of-the-art results on benchmarks like nuScenes, NAVSIM, and CARLA.

## 7 ACKNOWLEDGMENTS

This work was supported by National Science and Technology Major Project (2022ZD0116500). This work was also supported in part by the National Key R&D Program of China (No. 2022ZD0116500), the National Natural Science Foundation of China (No. U21B2042, No. 62320106010), and in part by the 2035 Innovation Program of CAS.

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

# A APPENDIX

## A.1 PREDICTING MULTIPLE FEATURES USING MULTIPLE INPUT FRAMES

**Predicting multiple future features** To better investigate the ability of our latent world model, we utilize the latent world model to predict multiple future frame latents, with the results presented in Table 8. We conduct this experiment using only the front-view camera to facilitate fast training. In detail, the future frame latents are predicted in an auto-regressive manner. For example, we first predicted the latent for 1.5 seconds into the future, then used this predicted latent to further predict the latent for 3 seconds into the future. The latent world model shares the same weights throughout this process. The results demonstrate that predicting multiple future latents further improves performance.

Table 8: **Predicting multiple future latents in an auto-regressive manner.**

| Predicted Future | L2 (m) ↓ | | | | Collision (%) ↓ | | | |
|---|---|---|---|---|---|---|---|---|
| | 1s | 2s | 3s | Avg. | 1s | 2s | 3s | Avg. |
| 1.5s | 0.34 | 0.69 | 1.17 | 0.73 | 0.12 | 0.22 | 0.63 | 0.32 |
| 1.5s → 3s | 0.31 | 0.65 | 1.12 | 0.69 | 0.11 | 0.19 | 0.57 | 0.29 |

**Predicting multiple future features using multiple input frames** Building on the previous section, we further conduct experiments to predict multiple future frame latents while incorporating multiple input frame latents to leverage temporal information more effectively. Specifically, we adopted a two-stage training paradigm for improved convergence, inspired by SOLOFusion (Park et al., 2022). In the first stage, we trained the model using single input frame latents for 12 epochs. This corresponds to the model denoted as "1.5s → 3s" in Table 8. In the second stage, we fine-tuned the model for an additional 6 epochs, now using two input frame latents, from 0s and 1.5s ago. The results are summarized in Table 9. The baseline (first row) represents the model fine-tuned using only single input frame latents. In contrast, the second row corresponds to the model fine-tuned with two input frame latents. The latter achieves significantly better performance. This highlights the crucial role of temporal information in autonomous driving.

Table 9: **Predicting future latents with multiple history frame inputs.**

| Predicted Future | Input Frames | L2 (m) ↓ | | | | Collision (%) ↓ | | | |
|---|---|---|---|---|---|---|---|---|---|
| | | 1s | 2s | 3s | Avg. | 1s | 2s | 3s | Avg. |
| 1.5s → 3s | 0s | 0.30 | 0.64 | 1.09 | 0.68 | 0.14 | 0.23 | 0.62 | 0.33 |
| 1.5s → 3s | -1.5s, 0s | 0.26 | 0.51 | 0.87 | 0.55 | 0.08 | 0.09 | 0.33 | 0.17 |

## A.2 MORE VISUALIZATION

In the appendix, we provide more visualization figures. We also provide a demo based on the CARLA simulator in the supplementary materials.

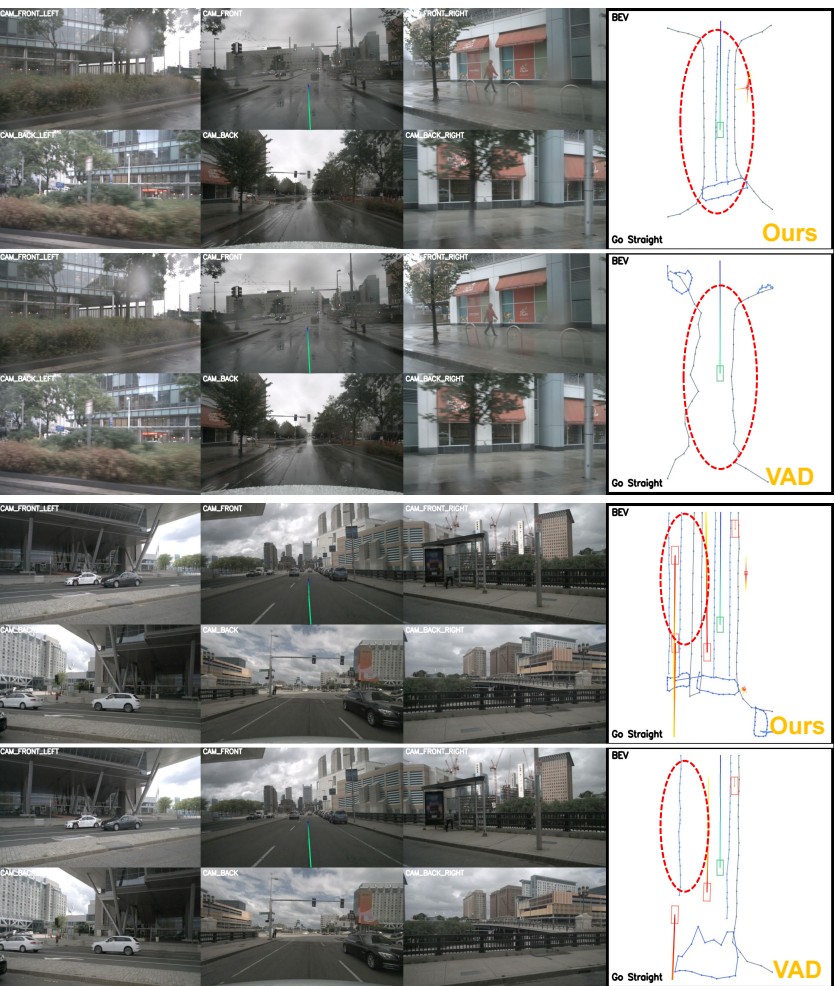

Figure 4: **Visualization.** As shown in the red circle, our map construction results are noticeably better than those of VAD.

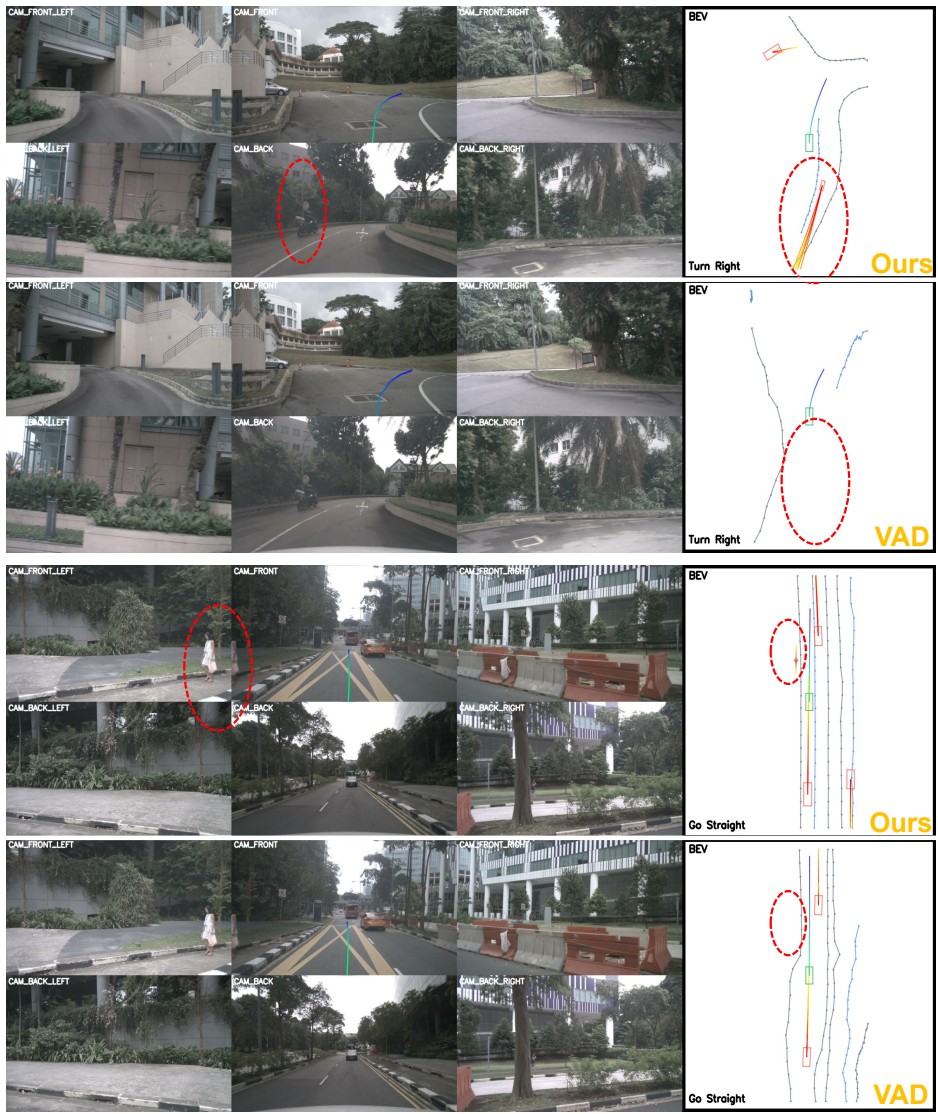

Figure 5: **Visualization.** As shown in the red circle, our agent motion prediction results are noticeably better than those of VAD.

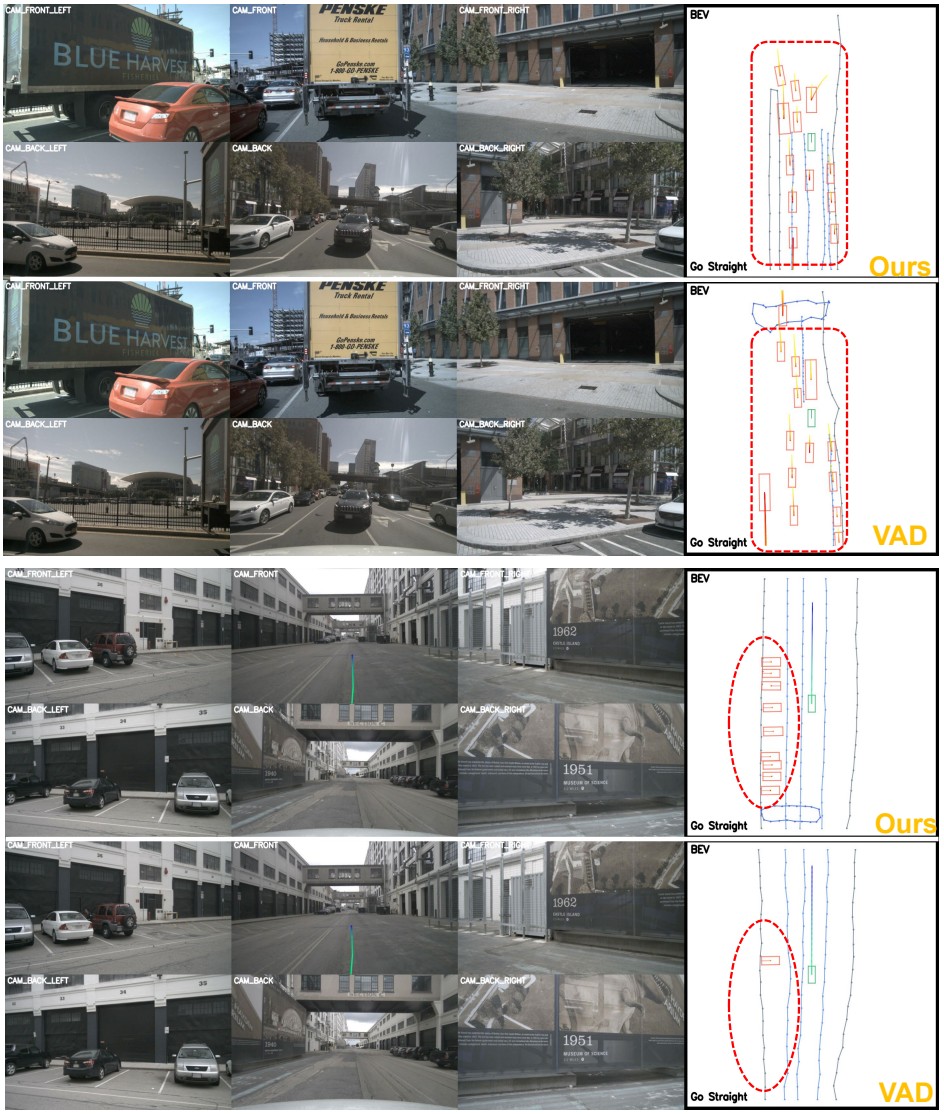

Figure 6: **Visualization.** As shown in the red circle, our map construction and agent motion prediction results are noticeably better than those of VAD, especially in heavily occluded and crowded conditions.

