# OpenReview forum: "Enhancing End-to-End Autonomous Driving with Latent World Model"
_ICLR.cc/2025/Conference — ICLR 2025 Poster_

### Official Review · Reviewer_z3s4 · 2024-10-21

**Soundness:** 2
**Presentation:** 3
**Contribution:** 2
**Rating:** 6
**Confidence:** 4

**Summary:**

This paper proposes a self-supervised learning approach, LAW, using the latent world model for end-to-end planning. The latent world model predicts future scene features based on the current scene features and predicted waypoints to learn better scene feature representation. This module is compatible with both perception-free and perception-based frameworks. The authors demonstrate that the proposed algorithm can beat existing baselines on three different benchmarks.

**Strengths:**

1. The proposed latent world model is a flexible plug-in module, which should be compatible with various end-to-end planning framework.
2. This paper conducts extensive experiments, showing the great performance on multiple benchmarks. The ablation study is comprehensive to cover different aspects of the latent world model.
3. The paper is well-written, very easy to understand and follow.

**Weaknesses:**

1. I think the idea of using self-supervised world model for autonomous driving is not novel, discussed in multiple prior works like GAIA-1, ADriver-I, and Drive-WM. The main difference of this work is the world model in latent space. However, it is not well-motivated why latent world model is better than world model in image space. For example, people can easily evaluate the image space world model by visualizing the prediction but it is hard to do the same thing for latent world model.

2. Following 1, the authors do not show any direct evaluation of world model future prediction itself. I notice that the performance gap of different time horizons are very small (Tab. 6) compared with the gain brought by world model (Tab. 4). Does the future prediction performance similar for different time horizons? If not, why their performances are similar? In my opinion, I do not think it is easy to predict the future if the time horizon is as long as 3s. Maybe the authors can at least provide the L2 error of the latent features on the validation set for different time horizons.

3. I think the settings in the experiment part are not fair. LAW uses Swin-T as the image backbone for nuScenes dataset. However, the baseline VAD and BEV-Planner are using ResNet-50, which is much weaker.

**Questions:**

Please consider replying to the points in the Weaknesses part.

---

> ### Author Response · Authors · 2024-11-24
> **Thanks and response to Reviewer z3s4 （1/2）**
>
> Thank you for your valuable and insightful comments. We deeply appreciate your input and have made every effort to address your concerns, as detailed below.
>
> ## Response to W1
> It seems there may be a misunderstanding regarding the primary **purpose** of our work. While prior studies, such as GAIA-1, ADriver-I, and Drive-WM, have indeed explored self-supervised world models for autonomous driving, their focus is primarily on **future image generation**. These works employ diffusion models to generate realistic future scene images, which are then utilized to facilitate reasonable driving decisions. In contrast, our work emphasizes using a world model to enhance end-to-end driving by **improving scene feature learning**.  To the best of our knowledge, we are the first to reveal that leveraging the world model to perform self-supervised future prediction tasks can help models learn better scene representation, thus improving end-to-end driving performance. While it is possible to test whether an image-based world model could similarly enhance scene feature learning, there are some issues.
>
> **Integrating an image-based world model into an end-to-end driving system is highly inefficient.** Existing image-based world models rely on diffusion processes to generate high-resolution images, which can take several seconds to produce a set of six-view images for a future scene. This process demands immense GPU memory and time. To address this, we propose a latent world model that predicts latent features instead of high-resolution images for future scenes. This approach is significantly more efficient in terms of time and memory usage. Our experiments demonstrate that the latent world model effectively enhances end-to-end driving performance.
>
> In summary, the novelty of our work lies in leveraging a world model to enhance feature representation learning, thus improving end-to-end driving performance. We greatly appreciate the reviewer's feedback and have revised the introduction section to clarify our motivation and prevent misunderstandings.
>
>
> ## Response to W2
> To address this concern, we conduct the experiments in the following table and find that **performance gains diminish over *longer* time horizons.** Specifically, we conducted an experiment predicting latent features 10 seconds into the future. The results indicate a significant decay in performance, which completely diminishes the gain brought by the world model. This experiment demonstrates that it is indeed challenging to predict the distant future.
>
> **However, a 3-second horizon is not long enough to eliminate the performance gains brought by the world model.** The reasons are discussed below. Low-level information in images can undergo dramatic changes within a relatively small horizon. However, high-level driving features tend to remain relatively stable over short horizons. For example, the road layout surrounding the ego vehicle is generally stable within a short horizon. To validate this, we provide the feature L2 distances. The L2 distances between the features at 1.5s, 3s, and 10s into the future and the current features are **0.0263**, **0.0420**, and **0.0752**, respectively. These results demonstrate that the feature changes within 3 seconds are significantly smaller than those within 10 seconds. As a result, we observe substantial performance gains over horizons such as 1.5s and 3s, which makes their performances seem similar.
>
> We thank the reviewer for highlighting this critical aspect of our work. We have incorporated these experiments and insights into the main paper as they provide a deeper understanding of future latent prediction.
>
> | Time Horizons | Feature L2 distance | L2 (m) Avg. | L2 (m) 1s | L2 (m) 2s | L2 (m) 3s | Collision (%) Avg. | Collision (%) 1s | Collision (%) 2s | Collision (%) 3s |
> | --- | --- | --- | --- | --- | --- | --- | --- | --- | --- |
> | w/o  world model | - | 0.71 | 0.32 | 0.67 | 1.14 | 0.41 | 0.20 | 0.30 | 0.73 |
> | 1.5s | 0.0263 | 0.58 | 0.26 | 0.54 | 0.93 | 0.25 | 0.14 | 0.17 | 0.45 |
> | 3s | 0.0420 | 0.63 | 0.28 | 0.59 | 1.01 | 0.27 | 0.13 | 0.20 | 0.48 |
> | 10s | 0.0752 | 0.72 | 0.33 | 0.67 | 1.16 | 0.43 | 0.24 | 0.28 | 0.76 |

---

> > ### Author Response · Authors · 2024-11-24
> > **Thanks and response to Reviewer z3s4 （2/2）**
> >
> > ## Response to W3
> > First, we would like to highlight that LAW uses ResNet-50 as the backbone of the perception-based framework. This already ensures a fair comparison. The results are shown in Table 1 of the main paper.
> >
> > Additionally, we provide results for LAW in the perception-free framework, also using ResNet-50 as the backbone, as presented in the following table.
> >
> > | Method | Backbone | L2 (m) 1s | L2 (m) 2s | L2 (m) 3s | L2 (m) Avg. | Collision (%) 1s | Collision (%) 2s | Collision (%) 3s | Collision (%) Avg. |
> > | --- | --- | --- | --- | --- | --- | --- | --- | --- | --- |
> > | VAD | ResNet-50 | 0.41 | 0.70 | 1.05 | 0.72 | 0.07 | 0.17 | 0.41 | 0.22 |
> > | LAW (perception-based) | ResNet-50 | 0.24 | 0.46 | 0.76 | 0.49 | 0.08 | 0.10 | 0.39 | 0.19 |
> > | LAW  (perception-free) | ResNet-50 | 0.29 | 0.62 | 1.10 | 0.67 | 0.14 | 0.27 | 0.74 | 0.38 |
> > | LAW  (perception-free) | Swin-Tiny | 0.26 | 0.57 | 1.01 | 0.61 | 0.14 | 0.21 | 0.54 | 0.30 |

---

> > > ### Comment · Reviewer_z3s4 · 2024-11-27
> > >
> > > Thanks for the detailed response. I think it can address my concerns, so I raise my score to 6.

---

> > > > ### Author Response · Authors · 2024-11-28
> > > > **Thanks for Reviewer z3s4's valuable feedback.**
> > > >
> > > > Thank you for your positive feedback and the higher score. Your suggestions have greatly refined the introduction of our work and have deepened our exploration of the temporal mechanisms.

---

### Official Review · Reviewer_g24j · 2024-11-02

**Soundness:** 3
**Presentation:** 3
**Contribution:** 3
**Rating:** 8
**Confidence:** 5

**Summary:**

This paper proposes the Latent World Model (LAW) to predict future features based on current features and ego trajectories, presenting a novel approach for end-to-end autonomous driving.

**Strengths:**

1. Introduction of the Latent World Model (LAW) to predict future scene latents from current scene latents and ego trajectories.

2. Demonstrated universality across various common autonomous driving paradigms, i.e.,  perception-free and perception-based frameworks.

3. Extensive experiments conducted on multiple benchmarks, achieving state-of-the-art performance on real-world open-loop datasets like nuScenes and simulator-based closed-loop CARLA benchmark.

**Weaknesses:**

See the Questions section.

**Questions:**

1. What is the shape of the Visual Latents, and can provide ablation studies?

2. Will LAW's use of the unique trajectory of the next frame as ground truth supervision limits the model's capabilities, given that trajectories are often not unique? Since the paper uses a simulation platform, could experiments with multi-trajectory supervision be provided?

3. LAW currently only uses the current frame latent to predict the next frame latent, while the introduction mentions that "using temporal data is crucial."

   3.1. Can experiments be conducted to predict multiple future frame latents?

   3.2. Can the model be trained to predict multiple future frame latents while taking in multiple input frame latents, leveraging temporal information more effectively?

4. Given that LAW currently predicts only the next frame latent based on the current frame latent, how does it handle tasks like predicting multiple future frames in nuScenes? Is this achieved through progressive prediction of the next frame latent?

---

> ### Author Response · Authors · 2024-11-24
> **Thanks and response to Reviewer g24j (1/2)**
>
> Thank you for your valuable and insightful comments. We deeply appreciate your input and have made every effort to address your concerns, as detailed below.
>
> ## Response to Q1
> We introduce the shape of Visual Latents as follows. For the perception-based framework, the Visual Latents need to support tasks such as detection and map construction, resulting in a relatively large shape. Specifically, the BEV feature map has dimensions of [25, 25, 256], where h = w = 25 and C = 256. Flattening this feature map results in Visual Latents of shape [625, 256]. In contrast, the perception-free framework does not require perception tasks, focusing on learning high-level planning features. Consequently, the shape of the Visual Latents is smaller. The shape is [36, 256]. With 6 views, each view contains 6 feature vectors, and each feature vector has 256 channels. We further conduct an ablation study on the number of Visual Latents, summarized in the table below. The results indicate that increasing the number of Visual Latents results in a higher L2 displacement error but a lower collision rate. We appreciate your insight into the importance of Visual Latent shapes.
>
> | #Visual Latents | L2 (m) 1s | L2 (m) 2s | L2 (m) 3s | L2 (m) Avg. | Collision (%) 1s | Collision (%) 2s | Collision (%) 3s | Collision (%) Avg. |
> | --- | --- | --- | --- | --- | --- | --- | --- | --- |
> | 36 | 0.26 | 0.57 | 1.01 | 0.61 | 0.14 | 0.21 | 0.54 | 0.30 |
> | 72 | 0.30 | 0.64 | 1.10 | 0.68 | 0.13 | 0.19 | 0.50 | 0.27 |
>
>
> ## Response to Q2
> Thank you for the insightful feedback. We considered conducting experiments with multi-trajectory supervision using the CARLA simulator. However, collecting future frames for multiple trajectories through the CARLA API was very time-consuming. This is due to CARLA's limited rendering efficiency, which results in slow image generation, especially when producing multiple sets of future frames for different trajectories. As a result, it becomes challenging to complete data collection, model training, and testing within a two-week timeframe. We sincerely appreciate the reviewer's valuable suggestion and plan to investigate this direction further in future work.
>
> Meanwhile, as an alternative, we can provide a similar experiment that involves predicting **multiple** future latents, as detailed in ''Response to Q3.1''. We believe that their impacts on performances will be similar, as both of them introduce **more supervision** to enhance feature learning.

---

> ### Author Response · Authors · 2024-11-24
> **Thanks and response to Reviewer g24j (2/2)**
>
> ## Response to Q3.1
> To address this concern, we utilize the latent world model to predict multiple future frame latents, with the results presented in the following tables. Due to the increased time required for extracting and predicting multiple future latents, we conduct this experiment using only the front-view camera to facilitate faster training. In detail, the future frame latents are predicted in an **auto-regressive** manner. For example, we first predicted the latent for 1.5 seconds into the future, then used this predicted latent to further predict the latent for 3 seconds into the future. The latent world model shares the same weights throughout this process. The results demonstrate that predicting multiple future latents further improves performance, as it introduces additional supervision. Thank you for your suggestion. We have incorporated these experiments into the revised paper.
>
> | Predicted Future | L2 (m) 1s | L2 (m) 2s | L2 (m) 3s | L2 (m) Avg. | Collision (%) 1s | Collision (%) 2s | Collision (%) 3s | Collision (%) Avg. |
> | --- | --- | --- | --- | --- | --- | --- | --- | --- |
> | 1.5s | 0.34 | 0.69 | 1.17 | 0.73 | 0.12 | 0.22 | 0.63 | 0.32 |
> | 1.5s, 3s (autoregressive) | 0.31 | 0.65 | 1.12 | 0.69 | 0.11 | 0.19 | 0.57 | 0.29 |
>
> ## Response to Q3.2
>
> Building on ''Response to Q3.1'', we conduct further experiments to predict multiple future frame latents while incorporating multiple input frame latents to leverage temporal information more effectively. Specifically, we adopted a two-stage training paradigm for improved convergence, inspired by SOLOFusion [1]. In the first stage, we trained the model using single input frame latents for 12 epochs. This corresponds to the model denoted as "1.5s, 3s (autoregressive)" in the table in ''Response to Q3.1''. In the second stage, we fine-tuned the model for an additional 6 epochs, now using two input frame latents, from the current and 1.5s ago. The results are summarized in the table below.
>
> The baseline (first row) represents the model fine-tuned using only a single current latent. In contrast, the second row corresponds to the model fine-tuned with two input frame latents. The latter achieves significantly better performance. This highlights the crucial role of temporal information in autonomous driving, as discussed in our introduction. Thank you for your insightful suggestions! We have incorporated these experiments into the revised paper.
>
> | Predicted Future | Input Frames | L2 (m) 1s | L2 (m) 2s | L2 (m) 3s | L2 (m) Avg. | Collision (%) 1s | Collision (%) 2s | Collision (%) 3s | Collision (%) Avg. |
> | --- | --- | --- | --- | --- | --- | --- | --- | --- | --- |
> | 1.5s, 3s | 0s | 0.30 | 0.64 | 1.09 | 0.68 | 0.14 | 0.23 | 0.62 | 0.33 |
> | 1.5s,  3s | -1.5s, 0s | 0.26 | 0.51 | 0.87 | 0.55 | 0.08 | 0.09 | 0.33 | 0.17 |
>
> ## Response to Q4
>
> Yes, as discussed in ''Response to Q3.1'', we employ an **auto-regressive** paradigm to predict multiple future latents. This approach involves step-by-step prediction, where each predicted latent is used as input for the subsequent step. The latent world model is used with shared weights throughout the process.
>
> [1]: Time Will Tell: New Outlooks and A Baseline for Temporal Multi-View 3D Object Detection

---

> > ### Comment · Reviewer_g24j · 2024-11-26
> >
> > Thank you for your insights and efforts. I have already raised my scores.

---

> > > ### Author Response · Authors · 2024-11-28
> > > **Thanks for Reviewer g24j's valuable feedback.**
> > >
> > > Thank you for your positive feedback and the increased score. Your suggestions on the temporal experiments have been incredibly insightful and have greatly enhanced the quality of our work. We look forward to further exploring this direction in the future.

---

### Official Review · Reviewer_AFWv · 2024-11-03

**Soundness:** 3
**Presentation:** 3
**Contribution:** 3
**Rating:** 8
**Confidence:** 4

**Summary:**

This paper introduces LAW, a self-supervised method that enhances end-to-end autonomous driving without the need for expensive annotations. It employs a latent world model to predict future latent features based on current data and actions, along with a view selection strategy to improve model efficiency. LAW outperforms state-of-the-art methods on both open-loop and closed-loop benchmarks.

**Strengths:**

The proposed LAW framework utilized a self-supervised method to significantly reduce the need for heavy annotation tasks, addressing the data scalability challenge of many existing methods. The detailed breakdowns of ablation studies, latency analyses, and visualizations provide readers with clear and comprehensive information to understand and reproduce the work.

**Weaknesses:**

The view selection strategy is a valuable insight to improve the efficiency of the method, but it adds complexity to the overall framework. Although there is only a minimal performance drop, it seems the view selection strategy hasn’t fully captured the informative scenes in driving scenarios. If there could be more discussion or analysis on what caused the performance drop, or how this issue could be mitigated with the Latent World Model, it would make the work more complete.

**Questions:**

Current View Selection Strategy Adaptability: Currently, the view selection strategy is designed for six cameras. If the number of cameras changes, how much effort is required to adjust the strategy? Is it possible to design the strategy to be adaptive to: 1) the total number of cameras, and 2) the selected number of cameras used at each time step—for example, using two cameras at  t=0  and three cameras at  t=1 ? In that case, how would you modify the selection reward?

Typographical Error in ‘Implementation Details’: On page 7 in the “Implementation Details” section, the last sentence starting with “For the closed-loop benchmark. And we use…” seems to contain a typographical error or is missing information.

Enhancing System with Additional Supervision: In Table 1, the ablation study shows the effectiveness of latent prediction. Additionally, what other forms of supervision could be added to further improve the system?

Performance with View Selection Strategy: In Tables 1 and 2, the open-loop and closed-loop tests are based on the default LAW, which does not use the View Selection strategy. If that’s the case, could you clarify the performance when the View Selection strategy is included?

Performance Discrepancy in Table 7: In Table 7, why is the performance using six views worse than using the Front + GT Views—for example, why is six-view performance worse than two-view performance? Could you explain the possible reasons?

Metrics at Other Time Horizons: In Table 4, the time horizons for latent prediction are only 0.5, 1.5, and 3.0 seconds. How does the metric change at other time horizons, such as 1 or 2 seconds?

---

> ### Author Response · Authors · 2024-11-24
> **Thanks and response to Reviewer AFWv (1/2)**
>
> Thank you for your valuable and insightful comments. We deeply appreciate your input and have made every effort to address your concerns, as detailed below.
>
> ## Response to Q1
>
> The view selection strategy can be adapted to different numbers of cameras using the following designs:
>
> 1. when the total number of cameras changes:
>
>     **We only need to adjust the number of selection queries when the total number of cameras changes.** The reasons are as follows. First, we do not need to change the main pipeline as both our feature extractor and latent world model are view-independent. View-independent means we extract each view’s feature and predict the future feature of this view independently. As a result, the overall framework is not affected by the change of the total number of cameras. Therefore, we only need to adjust the number of selection queries according to the total number of cameras. For example, we have 6 cameras now. We select the front camera and also select another camera out of five cameras, so the number of selection queries is C(5, 1)=5, where C(n, k) represents the number of combinations of selecting k items from n items. When we have 7 cameras, the number of selection queries will be C(6,1)=6.
>
> 2. when the selected number of cameras changes:
>
>     **We address this issue by increasing the number of selection queries to encompass all possible configurations of selected cameras.** For instance, with six cameras, there are 2^6 = 64 possible configurations, where each camera can either be selected or not. To account for all these configurations, we leverage 64 selection queries, with each query corresponding to a specific configuration. The corresponding 64 ground truth rewards can be generated using the original pipeline. By training the model to predict rewards for all these configurations, it can provide predicted rewards for any combination of selected cameras at each time step. This approach enables the framework to adapt seamlessly to scenarios where the number of selected cameras varies over time. Moreover, the framework can determine the optimal number of cameras to select for the next time step based on these predicted rewards.
>
> ## Response to Q2
> Thank you for pointing this out. The revised sentence is: “For the closed-loop benchmark, we use ….”
>
> ## Response to Q3
> To further enhance the system with additional supervision, we incorporate a detection task for bounding box supervision and a map construction task for map supervision into our framework. The results are in the following table. It shows that these additional supervision tasks (detection and map constructions) can further improve the system.
>
> | Supervision | L2 (m) 1s | L2 (m) 2s | L2 (m) 3s | L2 (m) Avg. | Collision (%) 1s | Collision (%) 2s | Collision (%) 3s | Collision (%) Avg. |
> | --- | --- | --- | --- | --- | --- | --- | --- | --- |
> | Latent Prediction | 0.26 | 0.57 | 1.01 | 0.61 | 0.14 | 0.21 | 0.54 | 0.30 |
> | Box + Map + Latent Prediciton | 0.24 | 0.46 | 0.76 | 0.49 | 0.08 | 0.10 | 0.39 | 0.19 |
>
> ## Response to Q4
> The performance when the view selection strategy is included is in Table 7 of the main paper. We also provide a copy of Table 7 as follows for your convenience. The second row shows the performance when the View Selection strategy is included.
>
> | Selected Views | L2 (m) 1s | L2 (m) 2s | L2 (m) 3s | L2 (m) Avg. | Collision (%) 1s | Collision (%) 2s | Collision (%) 3s | Collision (%) Avg. |
> | --- | --- | --- | --- | --- | --- | --- | --- | --- |
> | Front + a random view | 0.36 | 0.73 | 1.23 | 0.77 | 0.16 | 0.27 | 0.78 | 0.40 |
> | Front + predicted view | 0.30 | 0.64 | 1.10 | 0.68 | 0.16 | 0.25 | 0.72 | 0.38 |
> | Front + GT view | 0.28 | 0.56 | 0.97 | 0.60 | 0.15 | 0.22 | 0.61 | 0.33 |
> | Six views | 0.26 | 0.57 | 1.01 | 0.61 | 0.14 | 0.21 | 0.54 | 0.30 |
>
> ## Response to Q5
> There might be a misunderstanding here, as the settings for "Six Views" and "Front + GT Views" in the table represent two different setups. The "Front + GT Views" experiment leverages GT waypoint information as a **pilot study** to explore the performance upper bound, whereas the "Six Views" experiment does not utilize GT information. Specifically, the "Front + GT Views" experiment involves selecting the front camera and pairing it with one of the remaining five cameras, creating five combinations. Each combination generates a set of waypoints, and among these five sets, the one closest to the GT waypoints (based on their distances) is selected. This process inherently utilizes GT information, which explains its superior performance compared to the "Six Views" setting.

---

> > ### Author Response · Authors · 2024-11-24
> > **Thanks and response to Reviewer AFWv (2/2)**
> >
> > ## Response to Q6
> > The results at other time horizons are presented in the table below. We observe that performance improves as the time horizon extends from 0.5 s to 1.5 s. This improvement occurs because longer horizons provide richer future information for supervision, whereas shorter horizons offer limited new information.  However, we also note a performance decline when the time horizon extends from 1.5 s to 3 s. This decline can be attributed to the increased difficulty of accurately predicting distant futures as the time horizon grows. We appreciate your valuable feedback and will incorporate this table into the main paper, as it offers a more comprehensive understanding of future latent prediction tasks.
> >
> > | Time Horizons | L2 (m) 1s | L2 (m) 2s | L2 (m) 3s | L2 (m) Avg. | Collision (%) 1s | Collision (%) 2s | Collision (%) 3s | Collision (%) Avg. |
> > | --- | --- | --- | --- | --- | --- | --- | --- | --- |
> > | 0.5s | 0.26 | 0.57 | 1.01 | 0.61 | 0.14 | 0.21 | 0.54 | 0.30 |
> > | 1s | 0.26 | 0.53 | 0.93 | 0.57 | 0.12 | 0.15 | 0.46 | 0.24 |
> > | 1.5s | 0.26 | 0.54 | 0.93 | 0.58 | 0.14 | 0.17 | 0.45 | 0.25 |
> > | 2s | 0.28 | 0.56 | 0.97 | 0.60 | 0.10 | 0.17 | 0.52 | 0.26 |
> > | 3.0s | 0.28 | 0.59 | 1.01 | 0.63 | 0.13 | 0.20 | 0.48 | 0.27 |
> >
> > ## Response to Weakness
> > Here, we delve deeper into the reasons behind the performance drop observed when introducing the view-selection strategy. Additionally, we propose a potential solution by leveraging the Latent World Model.
> >
> > **Analysis of the performance drop.**
> > One primary reason for the performance drop is that the number of selected cameras is fixed. In complex scenarios that require input from multiple cameras to capture critical information, restricting the selection to a predefined number (e.g., two cameras) can lead to the omission of important details.
> >
> > **Mitigating this issue using the latent world model.**
> > To address this issue, we propose adaptively determining the number of cameras required for each scenario. The detailed implementation of utilizing a dynamic number of cameras is presented in the section “Response to Q1.” By leveraging future latent predictions from the latent world model, we can dynamically decide the optimal number of cameras needed to ensure comprehensive coverage of critical information. We sincerely thank the reviewer for the valuable suggestion to discuss this aspect, which significantly enhances the completeness of our work.

---

> > > ### Comment · Reviewer_AFWv · 2024-11-25
> > >
> > > Thank you to the authors for their detailed responses. The additional experiments largely address the questions and concerns raised about the paper, also the future work.Please update these futher analysis in your revised version. I will continue to follow your future work on scalability and efficiency in real-time implementations.

---

> > > > ### Author Response · Authors · 2024-11-28
> > > > **Thanks for Reviewer AFWv's valuable feedback.**
> > > >
> > > > Thank you for your thoughtful suggestions, which have significantly enhanced the quality of our paper. Moving forward, we are committed to further exploring scalability and efficiency, as these are key areas for the advancement of autonomous driving, particularly in real-time implementations.

---

### Official Review · Reviewer_MgUw · 2024-11-04

**Soundness:** 3
**Presentation:** 3
**Contribution:** 3
**Rating:** 6
**Confidence:** 5

**Summary:**

This paper introduces LAW (LAtent World model), a self-supervised learning approach for end-to-end autonomous driving. The key innovation is using a latent world model to predict future scene features based on current features and ego trajectories. The method can be integrated into both perception-free and perception-based frameworks, predicting either perspective-view features or BEV (Bird's Eye View) features respectively. The authors demonstrate state-of-the-art performance across multiple benchmarks including nuScenes, NAVSIM, and CARLA simulator.

**Strengths:**

1. Novel integration of world model concepts into end-to-end driving
2. Comprehensive experimental validation across multiple benchmarks. Demonstrates practical improvements in both closed and open-loop settings

**Weaknesses:**

1. Limited discussion of computational overhead - no analysis of inference time or model size. Autonomous driving systems must make decisions in real-time, typically requiring processing speeds of at least 10-20 Hz (decisions every 50-100ms). Without inference time analysis, it's unclear if LAW is applicable for real deployment on edge computing devices.
2. No discussion of robustness to adverse weather/lighting conditions. As in Appendix A.1, the augmentation is claimed to enhance the robustness, but there is no experiments to validate such ability of the model.
3. Missing references:
- [1] DriveVLM: The convergence of autonomous driving and large vision-language models
- [2] EMMA: End-to-End Multimodal Model for Autonomous Driving
- [3] VLP: Vision Language Planning for Autonomous Driving
- [4] OmniDrive: A Holistic LLM-Agent Framework for Autonomous Driving with 3D Perception, Reasoning and Planning

**Questions:**

1. What is the computational overhead of adding the latent world model? How does this impact real-time performance?
2. How does the method perform under challenging weather conditions or poor lighting? Are there specific failure modes?

---

> ### Author Response · Authors · 2024-11-24
> **Thanks and response to Reviewer MgUw**
>
> Thank you for your valuable and insightful comments. We deeply appreciate your input and have made every effort to address your concerns, as detailed below.
>
> ## Response to W1
> We provide the latency results in the following table. The latency of our model is tested on a single RTX 3090 GPU. For our perception-based framework, we adopt VAD-Tiny as the baseline. Since the latent world model is only utilized during training, the inference latency remains identical to VAD-Tiny. For the perception-free framework, our method achieves a latency of 51.2 ms, corresponding to ~20 FPS, which meets the requirements for real-time deployment on edge computing devices. Thank you for your insightful comment. We have incorporated the latency results into Table 1 in the revised version.
>
> | Method | Latency (ms) |
> | --- | --- |
> | UniAD | 555.6 |
> | VAD-Base | 224.3 |
> | VAD-Tiny | 59.5 |
> | LAW perception-based | 59.5 |
> | LAW perception-free | 51.2 |
>
> ## Response to W2
> We provide the results under various weather and lighting conditions in the table below. Our findings demonstrate that the model exhibits robustness to diverse weather and lighting scenarios. We sincerely thank the reviewer for encouraging us to explore this aspect, which makes our work more comprehensive. We have incorporated these results in the revised version.
>
> | Conditions | L2 (m) 1s | L2 (m) 2s | L2 (m) 3s | L2 (m) Avg. | Collision (%) 1s | Collision (%) 2s | Collision (%) 3s | Collision (%) Avg. |
> | --- | --- | --- | --- | --- | --- | --- | --- | --- |
> | All | 0.26 | 0.54 | 0.93 | 0.58 | 0.14 | 0.17 | 0.45 | 0.25 |
> | Day | 0.25 | 0.51 | 0.90 | 0.56 | 0.15 | 0.18 | 0.45 | 0.26 |
> | Night | 0.30 | 0.62 | 1.09 | 0.67 | 0.06 | 0.11 | 0.49 | 0.22 |
> | Sunny | 0.26 | 0.54 | 0.94 | 0.58 | 0.16 | 0.20 | 0.50 | 0.29 |
> | Rainy | 0.25 | 0.50 | 0.86 | 0.54 | 0.05 | 0.10 | 0.32 | 0.16 |
>
> ## Response to W3
> These references significantly enrich the depth of our work, and we have incorporated them into the related works section in the revised version. The corresponding sentences are as follows: DriveVLM [1] utilizes Vision-Language Models (VLMs) to improve comprehension of complex scenes and to optimize planning processes. EMMA [2] integrates multiple driving-related tasks into a unified language framework, employing task-specific prompts to generate the respective outputs. VLP [3] advances autonomous driving by enhancing the foundational memory of the source model and improving the contextual awareness of self-driving systems. OmniDrive [4] presents a comprehensive approach that ensures robust alignment between agent models and the requirements of 3D driving scenarios.
>
> [1] DriveVLM: The convergence of autonomous driving and large vision-language models
>
> [2] EMMA: End-to-End Multimodal Model for Autonomous Driving
>
> [3] VLP: Vision Language Planning for Autonomous Driving
>
> [4] OmniDrive: A Holistic LLM-Agent Framework for Autonomous Driving with 3D Perception, Reasoning and Planning

---

> > ### Comment · Reviewer_MgUw · 2024-11-26
> >
> > Thanks for the detailed response from the author. Both my concerns of latency and challenging conditions are properly addressed. I suggest authors to compare the results of challenging environments with other baselines in the future work. I will keep my positive rating.

---

> ### Author Response · Authors · 2024-11-28
> **Thanks for Reviewer MgUw's valuable feedback.**
>
> We sincerely appreciate your timely and insightful feedback. Your experiments on latency and challenging environments have been particularly constructive, and they have provided valuable guidance for improving our work. We are dedicated to further exploring challenging environments in our future work.

---

### Meta-Review · Area_Chair_6YnK · 2024-12-22

**Metareview:**

This paper proposed the LAtent World model (LAW), a method of self-supervised learning designed for end-to-end autonomous driving. The primary breakthrough lies in employing a latent world model to forecast upcoming scene characteristics, utilizing existing features and ego trajectories. The manuscript was reviewed by four experts in the field. The recommendations are (2 x "6: marginally above the acceptance threshold", 2 x "8: accept, good paper"). Based on the reviewers' feedback, the decision is to recommend the acceptance of the paper. The reviewers did raise some valuable concerns (especially additional and important experimental evaluations and ablation studied, needed comparisons with previous literature (clarification regarding technical insights), and together with further polishment of the manuscript) that should be addressed in the final camera-ready version of the paper. The authors are encouraged to make the necessary changes to the best of their ability.

**Additional Comments On Reviewer Discussion:**

Reviewers mainly hold the concern regarding extra additional and important experimental evaluations and ablation studies (Reviewers MgUw, AFWv, g24j, z3s4), needed comparisons with previous literature (Reviewer MgUw), and together with further polishment of the manuscript (Reviewers AFWv, z3s4). The authors address these concerns with detailed and extra experiments and commit to polishing the revised version further.

---

### Decision · Program_Chairs · 2025-01-22

Accept (Poster)